# A Rationally Designed H5 Hemagglutinin Subunit Vaccine Provides Broad-Spectrum Protection against Various H5Nx Highly Pathogenic Avian Influenza Viruses in Chickens

**DOI:** 10.3390/vaccines12080932

**Published:** 2024-08-21

**Authors:** Xuxiao Zhang, Fushou Zhang, Ning Chen, Xiaoping Cui, Xiaoqin Guo, Zhi Sun, Pengju Guo, Ming Liao, Xin Li

**Affiliations:** 1Boehringer Ingelheim Vetmedica (China) Co., Ltd., Taizhou 225300, China; xuxiao.zhang@boehringer-ingelheim.com (X.Z.); fushou.zhang@boehringer-ingelheim.com (F.Z.); xiaoqin.guo@boehringer-ingelheim.com (X.G.); zhi_1.sun@boehringer-ingelheim.com (Z.S.); tim.guo@boehringer-ingelheim.com (P.G.); 2National and Regional Joint Engineering Laboratory for Medicament of Zoonosis Prevention and Control, College of Veterinary Medicine, South China Agricultural University, Guangzhou 510642, China; 3Boehringer Ingelheim Animal Health USA Inc., 3239 Satellite Blvd, Duluth, GA 30096, USA; xiaoping.cui@boehringer-ingelheim.com; 4Key Laboratory for Prevention and Control of Avian Influenza and Other Major Poultry Diseases, Ministry of Agriculture and Rural Affairs, Institute of Animal Health, Guangdong Academy of Agricultural Sciences, Guangzhou 510640, China; 5College of Animal Science and Technology, Zhongkai University of Agricultural and Engineering, Guangzhou 510550, China

**Keywords:** avian influenza virus, H5 subtype, subunit vaccine, broad protection

## Abstract

The evolution of the H5 highly pathogenic avian influenza (HPAI) viruses has led to the emergence of distinct groups with genetically similar clusters of hemagglutinin (HA) sequences. In this study, a consensus H5 HA sequence was cloned into the baculovirus expression system. The HA protein was expressed in baculovirus-infected insect cells and utilized as the antigen for the production of an oil emulsion-based H5 avian influenza vaccine (rBacH5Con5Mut). Twenty-one-day-old SPF chickens were immunized with this vaccine and then challenged at 21 days post-vaccination with clade 2.3.2.1, clade 2.3.4.4, and clade 7.2 of H5 HPAI viruses. The sera of vaccinated chickens exhibited high hemagglutination inhibition (HI) titers against the rBacH5 vaccine antigen, while lower HI titers were observed against the different challenge virus H5 hemagglutinins. Furthermore, the rBacH5Con5Mut vaccine provided 100% protection from mortality and clinical signs. Virus isolation results showed that oropharyngeal and cloacal shedding was prevented in 100% of the vaccinated chickens when challenged with clade 2.3.2.1 and clade 2.3.4.4 H5 viruses. When the rBacH5Con5Mut vaccine candidate was administrated at one day of age, 100% protection was demonstrated against the challenge of clade 2.3.4.4 virus at three weeks of age, indicating the potential of this vaccine for hatchery vaccination. Overall, A single immunization of rBacH5Con5Mut vaccine candidate with a consensus HA antigen can protect chickens against different clades of H5 HPAI viruses throughout the rearing period of broiler chickens without a boost, thus fulfilling the criteria for an efficacious broad-spectrum H5 avian influenza vaccine.

## 1. Introduction

Highly pathogenic avian influenza (HPAI) virus infections cause significant economic losses to the poultry industry and pose a severe threat to public health. The H5 HPAI virus has spread among poultry and wild bird populations globally over the past two decades. These viruses have been categorized into 10 different clades based on the evolution of their hemagglutinin (HA) genes [1,2,3,4,5,6], and clade 2 has been further divided into different subclades. In China, clades 2.3.2, clade 2.3.4, and clade 7 viruses are responsible for most of disease outbreaks in domestic poultry. Most clade 2.3.2 viruses are clustered in Hong Kong, Henan, Inner Mongolia, and Qinghai [7]. Clade 2.3.4 viruses first emerged in the south and subsequently spread to the north. In contrast, clade 7 viruses were first isolated in northern China and spread from north to south into Hunan and Yunnan provinces [8].

Several countries have implemented a combination of culling and vaccination strategies to prevent and control the spread of H5 HPAI [5,9,10]. Since 2004, a series of recombinant inactivated vaccines have been developed and updated in China to match the antigenicity of the circulating strains of H5 viruses [9]. However, the rapid mutation rate of the H5 subtype HPAI viruses outpaces the time-consuming procedures of screening new candidate strains for the development of traditional inactivated whole-virus vaccines. Therefore, the existing vaccines might not provide adequate protection against the new emerging variants, posing a significant threat to the poultry industry in China [10]. In addition to the inactivated vaccines, a novel Newcastle disease virus (NDV)-vectored H5 avian influenza bivalent live vaccine has been used in chickens in China since 2006 [11]. In 2018, the first H5 DNA vaccine was approved [12].

Due to the high variations in HA sequences, especially within the neutralizing epitope regions, conventional HA-based H5 avian influenza vaccines do not appear to be protective against heterologous strains or phylogenetically variant clades of H5 avian influenza virus. Therefore, there is an urgent need to develop an effective, broadly cross-protective, and safe H5 avian influenza vaccine. A strategy to exploit a cocktail of antigenically different H5 virus strains has been tried to provide broad protection [13]. Other strategies are based on choosing immunogens from conserved regions to provide broad protection against different influenza viruses, such as HA stalk domain, highly conserved M2 ectodomain (M2e) [12], HA fusion domain (HFD) [13], T-cell epitope of nucleoprotein (TNP) [14], and HA α-helix domain (HαD) [15,16]. A novel “mosaic” approach which used an algorithm to select and recombine potential 9mer to 12mer “chunks” into a full-length protein in an attempt to preserve naturally occurring T-cell epitopes was introduced previously [17]. This strategy has been applied to generate a modified vaccinia Ankara-vectored vaccine expressing a mosaic H5 HA [17].

Recent studies have shown that highly conserved regions in the HA protein can induce antibodies that provide subtype-independent protection [18,19,20]. Consensus sequences, which are obtained by aligning a population of sequences and selecting the most common residue at each position, are another promising strategy for influenza vaccine development that can elicit cross-reactive immune responses [6,21,22,23]. There are reports of experimental vaccines with consensus sequence that can effectively protect mice [24,25,26], ferrets [26], and chickens [6] against H5N1 infection. In this study, an insect-cell-derived single consensus H5 HA antigen-based subunit vaccine (rBacH5Con5Mut) was developed and two mutations were introduced to further increase its antigenicity. This subunit vaccine can provide satisfactory protection against the lethal challenge of different clades of the H5 HPAI viruses.

## 2. Materials and Methods

### 2.1. Ethics Statement

All animal experiments were carried out in animal biosafety level 3 (ABSL-3) facilities in compliance with the protocols of the Biosafety Committee of the ABSL-3 Laboratory of the South China Agriculture University (SCAU, CNAS BL0011). All animal experiments were reviewed and approved by the Institutional Animal Care and Use Committee of the South China Agriculture University and were carried out in accordance with the approved guidelines (2017A002).

### 2.2. Cells and Viruses

The SF+ cells were maintained as suspension cultures in EX-CELL 420 insect-serum-free medium (Sigma, St. Louis, MO, USA) at 27 ± 2 °C, and Sf9 cells were maintained as adherent cultures in SF900-III medium (Gibco, Waltham, MA, USA) at 27 ± 2 °C. The 293T cells were purchased from ATCC, Cat: PTA-5077. Madin–Darby canine kidney (MDCK) cells were imported from Boehringer Ingelheim Vetmedica Inc., Duluth, GA, USA. The 293T and MDCK cells were maintained as adherent cultures in DMEM medium (Gibco, Waltham, MA, USA) with 10% FBS (Gibco, Waltham, MA, USA) at 37 °C.

The rBacH5Con5Mut virus stock was produced in SF+ cells. Briefly, the SF+ cells were infected with the virus at a low multiplicity of infection (MOI) of ≤0.01. The clarified supernatant was harvested at four days post-infection and stored at −80 °C. Three H5 HPAI viruses originating from clade 2.3.2.1 (A/Duck/Guangdong/383/2008), clade 2.3.4.4 (A/Goose/Guangdong/079/2013), and clade 7.2 (A/Duck/Shandong/147/2013) were isolated from the mainland of China between 2008 and 2013 and were characterized as the challenge strains by SCAU.

### 2.3. Antigen Design and Recombinant Baculovirus Production

The complete open reading frame (ORF) of the consensus HA sequence was obtained by aligning the 150 avian-origin complete H5 subtype HA sequences (isolated from 2005 to 2013 in China) in GenBank using the Vector NTI software. The highest frequency of amino acid in each position was selected to generate the consensus HA. The additional mutations including S120N and S223N were introduced into the selected HA gene as previously described [27]. This HA (rBacH5Con5Mut) ORF was then codon-optimized according to the codon usage frequency of *Spodoptera frugiperda* (*Sf*) cells by GeneScript Co., Ltd. (Nanjing, Jiangsu, China). The optimized rBacH5Con5Mut ORF was cloned into the pVL1393 vector plasmid. To generate the recombinant baculovirus (rBVs), the plasmid and linearized baculovirus genomic DNA (SapphireTM, baculovirus DNA and transfection kit, Orbigen, San Diego, CA, USA) were co-transfected into Sf9 insect cells according to the manufacturer’s instructions. Single rBV clones were obtained by plaque assay.

### 2.4. Phylogenetic Analysis

A total of 67 HA gene sequences analyzed in this study were downloaded from GenBank (http://www.ncbi.nlm.nih.gov/genomes/FLU/ (accessed on 26 August 2021)) and the Global Initiative on Sharing Avian Influenza Data database (https:www.gisaid.org (accessed on 26 August 2021)) under accession numbers ABJ96755, ABJ96964, ABJ96695, ABJ96774, ABO14790, ADG59062, AKI82312, ACH85498, AEO89172, ADG59077, AJK00229, AEO89235, AEO89246, ACB70691, ADF83650, AJK00177, ACJ15273, ACJ15251, ACJ15262, ACH85399, ACB70724, ACB70790, ABY19417, ACB70757, ACY80677, ACB70713, ACB70801, ACB70823, ABE97603, ADM88571, ACY80678, ACY80658, ACY80684, ACB70746, EPI1639975, EPI1639991, EPI1639983, EPI1639959, EPI1639967, EPI1639951, EPI1639935, EPI1639943, EPI1639887, EPI1639879, EPI1639863, EPI1639855, EPI1639847, EPI1639839, EPI1639871, EPI1639831, EPI1639823, EPI1639815, EPI657466, EPI749825, EPI589366, EPI1639919, EPI561660, EPI1366565, EPI930807, EPI1060783, EPI1366563, EPI1366564, and EPI1426910. HA sequences were aligned, and neighbor-joining (NJ) phylogenies were estimated using MEGA 7 software. NJ methods were used to perform system reconstruction and to calculate confidence values to estimate the number of repetitions (1000).

### 2.5. Western Blot

Because the SF+ cell is a suspension cell culture, the whole-cell suspension samples included both the culture supernatant and cells were directly mixed with loading buffer (Invitrogen, Waltham, MA, USA) containing lithium dodecyl sulfate at pH 8.5 with SERVA Blue G250 and phenol red (with or without reducing reagent (Invitrogen, Waltham, MA, USA, containing 500 mM dithiothreitol)) and denatured by heating for 5 min at 95 °C. The proteins were separated by denaturing polyacrylamide gel electrophoresis and then transferred to polyvinylidene fluoride membranes. After blocking with 5% skim milk in phosphate-buffered saline with 0.05% tween (PBST) for 30 min, the membranes were probed with mouse anti-H5N1 HA monoclonal antibody (Sino Biological, Beijing, China), and then with goat anti-mouse IgG-HRP (BIO-RAD, Hercules, CA, USA). The positive bands were visualized using the Supersignal West Femto Maximum Sensitivity Substrate (Thermo Scientific, Waltham, MA, USA).

### 2.6. Indirect Immunofluorescence Assay

To determine the expression of the HA protein and the baculovirus envelope glycoprotein 64 (Gp64), Sf9 cells were grown in 24-well plates and infected with the recombinant virus at an MOI of 1. At 5 days post-infection, the cells were fixed with 50% methanol and 50% acetone mixture for 15 min. The cells were incubated at 37 °C for 1 h with a rabbit polyclonal antibody (pAb) directed against the H5 HA protein (Sino Biological, Beijing, China) and mouse monoclonal antibody (mAb) directed against baculovirus envelope Gp64 protein (eBioscience, San Diego, CA, USA). The cells were then washed three times with PBS and incubated at 37 °C for 1 h with Alexa Fluor 594-conjugated donkey anti-mouse IgG (H+L) secondary antibody (Invitrogen, Carlsbad, CA, USA) and Alexa Fluor 488 donkey anti-rabbit IgG (H+L) secondary antibody (Invitrogen, Carlsbad, CA, USA). After incubation with the secondary antibody, the cells were washed three times with PBS and the cells were examined using a Leica fluorescence microscope.

### 2.7. Hemagglutination (HA) Assay

rBacH5Con5Mut-infected SF+ cell suspension or allantoic fluid samples (25 μL) were serially diluted 2-fold by phosphate-buffered saline (PBS) (Gibco, Waltham, MA, USA) from 1:2 to 1:2048 in 96-well V-bottom plates. In each well, 25 μL PBS was added in a 25 μL volume of the diluted samples, and then 25 μL 1% chicken red blood cells (RBCs) were added to each well. The plates were gently mixed and incubated at room temperature for 45 min. The highest dilution that showed complete hemagglutination was recorded as the HA titer for individual sample.

### 2.8. Preparation of the Vaccines

To prepare the vaccine antigen, SF+ cells were infected with the above-described rBacH5Con5Mut virus stock at an MOI of 0.1 and harvested six days later. The clarified whole-cell suspension was incubated with Binary Ethylenimine (BEI, final concentration of 10 mM) at 37 °C for 24 h to inactivate the baculovirus. Then the BEI was neutralized by mixing with NaS_2_O_3_·5H_2_O at 37 °C for 1 h. The inactivated samples were mixed with water-in-oil adjuvant (30% vaccine antigen/70% oil adjuvant, *v*/*v*) and emulsified to formulate the oil-adjuvanted inactivated vaccine. The content of HA protein in the final vaccine was measured by hemagglutination assay before emulsion.

The commercial trivalent inactivated whole-virus-based mineral oil emulsion Re-6-7-8 vaccine was obtained from Weike Biotechnology Co., Ltd., Harbin, China. The six internal genes of these recombinant vaccine strains were derived from A/Puerto Rico/8/1934 virus (PR8, H1N1). The HA and NA genes of Re-6-7-8 were derived from the strains of different subclades of H5 wild-type viruses: Re6 from A/duck/Guangdong/S1322/2006 (clade 2.3.2) [28], Re7 from A/Chicken/Liaoning/S4092/2011 (clade 7.2) [29], and Re8 from A/chicken/Guizhou/4/13 (clade 2.3.4.4) [30].

### 2.9. Animal Study

#### 2.9.1. Immunization and Challenge

Groups of 10 twenty-one-day-old or one-day-old White Leghorn SPF chickens were injected subcutaneously with 0.3 mL of oil-adjuvanted vaccines containing 500 HAU of HA antigens or 0.3 mL of commercial trivalent inactivated whole-virus-based mineral oil emulsion Re-6-7-8 vaccine. Groups of 10 chickens were injected subcutaneously with 0.3 mL of phosphate-buffered saline (PBS) as a control. Three weeks after immunization, serum samples were collected for antibody detection, and the birds were challenged intranasally with 10^6^ egg infectious dose 50 (EID_50_) of the challenge virus strain including A/Goose/Guangdong/079/2013 (H5N1), A/Duck/Guangdong/383/2008 (H5N1), and A/Duck/Shandong/147/2013 (H5N2). Oropharyngeal and cloacal swabs were collected at five days post-challenge for virus isolation. All birds were observed for signs of disease (such as loss of appetite, lack of energy, purple comb, hemorrhage of foot scale, palpebral edema, and neurological disorders (twisting the neck) or death) over a period of 14 days after the challenge and the remaining chickens were then humanely euthanized. For HPAI, the chickens in the challenge-only group died within 48 h, with a very short period of time showing clinical symptoms, and thus in general required no humane interventions such as euthanasia For the vaccination/challenge groups, we closely monitored animals and were always prepared to humanely euthanize the animals if any sign of suffering was shown. The survival rate is calculated by the number of surviving chickens divided by the number of total chickens over 14 days after challenge. The survival rate curves were generated with GraphPad Prism 10 (GraphPad Software Inc., San Diego, CA, USA).

#### 2.9.2. Virus Isolation

To assess virus shedding after the challenge, the collected cotton swab samples were processed and inoculated in SPF chicken embryos for virus isolation. Briefly, the cotton swabs were taken into labeled tubes pre-loaded with 1 mL PBS containing 2000 units of Pen–Strep. The samples were centrifuged at 3000× *g* for 10 min and the supernatants were transferred to fresh tubes and stored at −80 °C. Each sample was inoculated into the allantoic cavity of three 9- to 11-day-old SPF chicken embryos (0.2 mL per embryo), which were then incubated at 37 °C for 72 h. The embryos that were not viable within 24 h post-inoculation were discarded. Allantoic fluid was collected from each embryo, and HA activity was measured.

#### 2.9.3. Hemagglutination Inhibition (HI) Assay

Serum samples were collected and inactivated at 56 °C for 30 min. The antigen used for HI assay is either BEI-inactivated rBacH5Con5Mut or formaldehyde-inactivated challenge virus antigen. The serum samples were serially diluted 2-fold by phosphate-buffered saline (PBS) in a 96-well V-bottom plate, mixed with an equal volume of antigen (4 HA units), and incubated at room temperature for 30 min. After adding 1% chicken RBCs, the plates were gently mixed and incubated for 45 min. The HI titer was calculated as the highest dilution of serum causing complete inhibition of hemagglutination.

#### 2.9.4. Generation of Pseudovirus Bearing HA and NA of H5N1 Avian Influenza Virus

H5HA and N1NA genes derived from A/Goose/Guangdong/079/2013 were codon-optimized for the mammalian expression system. The 293T cells were seeded into 6-well plates and co-transfected with 6 µg plasmid encoding env-defective, luciferase-expressing HIV-1 (pNL4-3.luc.RE), 1.6 µg H5HA-pVAX1, and 1.6 ug N1NA-pVAX1. The medium was removed and replaced with fresh DMEM 4 h later. Supernatants containing the pseudovirus particles were harvested 48 h post-transfection and used for single-cycle infection. VSV-G pseudovirus was generated as a control by co-transfecting plasmids encoding vesicular stomatitis virus G protein (VSV-G-pcDNA3.1) with pNL4-3.luc.RE plasmid. For virus titration, MDCK cells were infected with 200 µL of each pseudovirus in 24-well culture plates, and the medium was replaced with fresh medium after 24 h. The infected cells were incubated for 48 h and the luciferase activity was detected. The pseudovirus titer was calculated in terms of relative luciferase units (RLUs) tested by Promega GloMax Navigator.

#### 2.9.5. HA Pseudovirus Neutralization Assay

MDCK cells were seeded in 96-well plates at the density of 5000 cells/well in 100 µL DMEM one day before the test. To prepare the virus samples, 10^4^ RLUs of H5N1 pseudo virus were incubated with 2-fold serially diluted sera (starting dilution 1:5) for 1 h at 37 °C (in a CO_2_ incubator) in 160 µL antibiotic-supplemented serum-free MEM. The MDCK cells were washed with PBS and incubated with 150 µL of the neutralized sample per well for 24 h. The medium was discarded and 150 µL fresh DMEM with 10% FBS was added to each well, and the cells were incubated for 48 h. The luciferase assay was performed by adding the Bright-Glo Luciferase substrate (Promega, Madison, WI, USA). The neutralization of HA pseudovirus was calculated and expressed in terms of the half maximal inhibitory concentration (IC50) calculated using GraphPad Prism 10.

## 3. Results

### 3.1. Characterization of Consensus H5Nx HA Expressed by Recombinant Baculovirus

The consensus H5 HA sequence was obtained by aligning the 150 avian-origin complete H5 subtype HA sequences (Figure 1a). Phylogenetic analysis results showed that this consensus H5 HA gene belonged to clade 2.3.4.1 (Figure 1b). S120N and S223N mutations were introduced into the H5 HA consensus sequence to increase the antigenicity. Structural modeling showed that these two mutations were both located in the globular head region (Figure 1c). HA expression in rBacH5Con5Mut-infected SF+ cells was confirmed by Western blot, dual IFA, and HA assays (Figure 1d–f). Bands corresponding to HA0 and HA1 were detected at around 65 kDa and 50 kDa, respectively, in the reducing condition. HA2 was not detected because the monoclonal antibody cannot bind with HA2. The presence of the HA1 band can be attributed to the partial cleavage of HA0 in the insect cells [31]. Furthermore, dimer (~130 kDa) and trimer (~195 kDa) forms of HA0 as well as the oligomer form with high molecular weight were detected in the non-reducing condition. SF+ cells infected with wild-type baculovirus were set as a mock control in this study, and there was no band detected in the control group which indicated that there was no nonspecific reaction of the H5 monoclonal antibody. The HA titer was 9 log2 at 5 days post-infection. Dual IFA assay also confirmed the expression of H5 HA. Moreover, the expression of Gp64 and HA overlapped in every plaque completely, which indicated that the rBacH5Con5Mut virus was a pure virus after plaque purification. Taken together, these results confirmed the successful expression of consensus H5 HA in rBacH5Con5Mut-infected SF+ cells. The expressed HA can be recognized by a H5 HA monoclonal antibody and elicits hemagglutination activity.

### 3.2. Genetic Stability of rBacH5Con5Mut Virus

The rBacH5Con5Mut virus was subjected to plaque purification and its monoclonality was verified according to the methods described above. To check the genetic stability of rBacH5Con5Mut, the virus was continuously passaged on SF+ cells with 10 passages to P17, and the H5 HA gene was analyzed by PCR and sequencing. The dual IFA assay confirmed the expression of H5 HA protein and the purity of the recombinant virus (Figure 2a). As shown in Figure 2b, there were no deletions in the H5 HA segment and the gene sequencing of the PCR production result showed that there were no mutations in the H5 HA gene when passaged to P17. Furthermore, the rBacH5Con5Mut virus from P7, P12, and P17 showed no significant differences in terms of H5 HA expression level or HA activity (Figure 2c,d). These findings indicated that the designed consensus H5 HA gene was stable in recombinant baculovirus for up to 17 passages.

### 3.3. A Single Immunization with rBacH5Con5Mut Vaccine Prevented Infection from 2.3.4.4 Clade H5N1 Virus

To test the efficacy of the rBacH5Con5Mut vaccine against widely circulating poultry HPAI strains in China, 21-day-old chickens were vaccinated with the rBacH5Con5Mut vaccine and then challenged with the A/Goose/Guangdong/079/2013 (H5N1, 14079) strain, belonging to clade 2.3.4.4, after 21 days post-immunization. In addition, a positive control group immunized with the Re-6-7-8 commercial vaccine and a PBS-treated mock control group were included in the study for comparison. Serum samples were collected at 21 days post-vaccination (dpv) from the different groups for serological tests.

As shown in Table 1, rBacH5Con5Mut induced an HI geometric mean titer (GMT) of 9.1 log2 against the recombinant H5 (rBacH5) antigen at 21 dpv. The highest HI titer of single sera was 10 log2, while the lowest HI titer was 8 log2. The average HI titer against the standard antigen of the challenge virus strain was 3.6 log2. In contrast, the HI titer of the sera of PBS-treated controls was less than 4 log2 for rBacH5 antigen and 0 log2 for the challenge virus standard antigen. Furthermore, the HI titers of the sera from Re-6-7-8 vaccination group were 6.5 log2 and 8.0 log2 against rBacH5 antigen and challenge strain, respectively.

The chickens were challenged at 21 dpv with 10^6^ EID_50_ of the 14079 virus. Clinical signs and mortality were monitored for 14 days. All PBS-treated chickens died within two days following the challenge. However, none of the chickens vaccinated with rBacH5con5Mut or Re-6-7-8 vaccine died or showed any clinical signs after infection, indicating that both vaccines provided 100% protection from mortality and morbidity (Figure 3a). Virus shedding was tested at 5 dpc by isolating the virus from both oropharyngeal and cloacal swab samples (Table 1). No shedding was observed from the cloacal or oropharyngeal samples of the rBacH5con5Mut and Re-6-7-8 vaccination groups. Taken together, these results clearly indicated this rBacH5Con5Mut subunit vaccine can protect against morbidity, mortality, and virus shedding in response to a lethal challenge of the highly pathogenic clade 2.3.4.4 H5N1 virus.

### 3.4. Pseudovirus-Based Neutralization Assay

In the current experiments, we found that even low HI antibody titers against challenge viruses in the rBacH5Con5Mut-vaccinated chickens could provide full protection. The pseudovirus-based neuralization assay was applied to further evaluate the immunogenicity of our vaccine candidate. The H5N1 HPAI virus must be handled in biosafety level 3 containment laboratories. To obviate this issue, one possible alternative is to use the pseudovirus to replace wild-type viruses. The pseudovirus lacks certain gene sequences of the parent virulent virus, and therefore can be handled in biosafety level 2 laboratories. To this end, we performed the H5N1 (14079) pseudovirus-based neutralization (pVN) test to detect serum neutralization antibody levels against influenza H5N1 (14079). The serum antibody titers were measured at 21 dpv. The average pVN antibody titers were 151 and 3241, respectively, in rBacH5Con5Mut vaccination and Re-6-7-8 vaccination groups (Figure 3b). Although the pVN titer is lower in the rBacH5Con5Mut vaccination group, it still can provide 100% protection against 14079 virus challenge.

### 3.5. The rBacH5Con5Mut Vaccine Was Cross-Protective against Clade 2.3.2.1 H5N1 and Clade 7.2 H5N2 Viruses

We also evaluated the efficacy of rBacH5Con5Mut against H5N1 A/Duck/Guangdong/383/2008 (clade 2.3.2.1) and H5N2 A/Duck/Shandong/147/2013 (clade 7.2) H5 viruses, given that these strains are widely circulating in China. The Re-6-7-8 vaccine-immunized and PBS-treated groups were included as controls, and the experiment was performed as described in the previous section.

In the efficacy study challenged with A/Duck/Guangdong/383/2008 (H5N1), the HI test result showed that the immune response to the rBacH5 antigen and the challenge virus on day 21 post-immunization was consistent with the response recorded for the vaccine batch in the previous experiment (Table 1). The average HI titers of the sera from the unvaccinated controls were 0 log2 and 3.3 log2 against the challenge virus and rBacH5 antigen, respectively. The average HI titers induced by rBacH5Con5Mut against the rBacH5 antigen and challenge virus were 8.0 log2 and 2.1 log2, respectively. In the Re-6-7-8 vaccination group, HI titers of 5.8 log2 and 5.7 log2 were recorded against the challenge virus and rBacH5 antigen, respectively. Infection with the clade 2.3.2.1 virus killed all chickens in PBS control group by day 4 post-infection. In contrast, all vaccinated chickens survived after the virus challenge without showing any clinical signs (Figure 4a). Furthermore, the virus isolation test showed that vaccination with rBacH5Con5Mut and Re-6-7-8 prevented both cloacal shedding and oropharyngeal shedding of the clade 2.3.2.1 virus at 5 dpc (Table 1).

Next, we evaluated the efficacy of rBacH5Con5Mut against the A/Duck/Shandong/147/2013 (H5N2) virus, which belongs to clade 7.2. Serological testing results showed that the HI titers of the unvaccinated controls against the challenge virus and rBacH5 antigen were 0 log2 and 3.3 log2, respectively. On the other hand, the HI titers of the sera from the rBacH5Con5Mut vaccination group were 8.7 log2 and 1.6 log2 against the vaccine strain and challenge strain, respectively, while those of the Re-6-7-8 vaccination group were 6.2 log2 and 6.3 log2. In the unvaccinated group, nine chickens died within 6 dpc, and one bird was sick starting at 4 dpc and died at 14 dpc (Figure 4b). All rBacH5Con5Mut-vaccinated chickens survived the challenge infection without showing any clinical signs during the observation period (Figure 4b). Virus isolation tests showed oropharyngeal shedding for one chicken and cloacal shedding for six chickens at 5 dpc. Likewise, virus shedding was observed in the oropharyngeal swab of one chicken from the Re-6-7-8 vaccination group (Table 1).

In summary, these results suggested that the rBacH5Con5Mut vaccine can provide cross-protection against the lethal challenge of clade 2.3.2.1 and clade 7.2 H5 viruses and prevent shedding of the clade 2.3.2.1 prevalent strain. Overall, this novel vaccine candidate exhibits a broad protection profile against HPAI viruses of different clades.

### 3.6. Protective Efficacy of rBacH5Con5Mut Vaccine Delivered to One-Day-Old SPF Chickens against 2.3.4.4 Clade H5N1 Virus

Hatchery immunization is crucial to prevent virus outbreaks in poultry farms. To determine whether rBacH5Con5Mut can be used in hatchery, we immunized a group of one-day-old SPF chickens with 0.3 mL of the rBacH5Con5Mut vaccine product and challenged them three weeks later with 10^6^ EID50 of 14079 H5N1 strain. Chickens injected with PBS were included as the mock control group. Serum samples were collected three weeks post-immunization, and the titers of serum antibodies were measured against rBacH5 and 14079 challenge virus antigens by the HI assay. At 3 weeks post-immunization, all the chickens immunized with the rBacH5Con5Mut vaccine mounted a positive HI antibody response against rBacH5 and 14079 standard antigens with mean titers of 9.1 log2 and 2.5 log2, respectively (Table 2). In contrast, no HI titers were detected in the non-immunized chickens during the entire observation period. Thus, the rBacH5Con5Mut vaccine candidate induced a positive HI antibody response in one-day-old chickens.

After lethal challenge, all chickens were monitored for 14 days for clinical signs and mortality. The non-immunized chickens showed the typical clinical signs, such as purple comb, hemorrhage of foot scale, and palpebral edema, and all birds died within two days, suggesting that 14079 virus infection is highly lethal to these chickens. However, none of the chickens immunized with the rBacH5Con5Mut vaccine showed any clinical symptoms during the observation period, and the survival rate was 100% (Figure 5).

To monitor virus shedding, the oropharyngeal and cloacal swabs were collected at 5 dpc for virus isolation. No shedding could be detected in the immunized group (Table 2). Taken together, these results indicated that the rBacH5Con5Mut vaccine provided complete protection against a lethal challenge of H5N1 virus and virus shedding when administrated to one-day-old SPF chickens.

## 4. Discussion

In this study, we developed a consensus H5 HA sequence-based subunit H5 avian influenza vaccine (rBacH5Con5Mut) and comprehensively analyzed its immunogenicity and efficacy in chickens on the basis of serum HI antibody titers, pseudovirus VN antibodies, and cross-protection against lethal challenge of three heterologous clades (2.3.4.4, 2.3.2.1 and 7.2) of H5 HPAI viruses. The vaccine based on a rationally designed consensus H5 HA antigen can induce broad-spectrum immunity against different clades of H5 avian influenza viruses. By encoding the most conserved residues found in a selected population, the consensus sequences can effectively target multiple clade viruses with different mutation patterns. In contrast, specific virus strain sequence-based subunit vaccines provide only strain-restricted immunity. Consensus sequence-based vaccines have been extensively investigated for eliciting broadly reactive immune responses against different pathogens, including influenza virus [21,32,33], chikungunya virus [34], hepatitis B virus [35], hepatitis C virus [36], HIV-1 [37], and Middle East respiratory virus [38].

The entire ORF of the consensus HA sequence was obtained after alignment from 150 H5 strains isolated between 2005 and 2013. Phylogenetic analysis results showed that this consensus H5 HA belonged to clade 2.3.4.1. The Ser120Asn mutation was added to match the HA sequences of H5N1 HPAI viruses isolated from both humans and poultry in 2003 in order to develop a vaccine against potential pandemics of H5 HPAI viruses. Furthermore, serine at amino acid residue 223 was substituted with asparagine in order to enhance the antigenicity of the vaccine candidate in chickens [39]. The replacement of Ser with Asn will not introduce an additional N-glycosylation site based on the online glycosylation prediction result. It needs to be noted that the Re-6-7-8 commercial vaccine is a trivalent product and produced by using three virus antigens while our rBacH5Con5Mut vaccine is generated by using a single HA antigen. Despite using only a single HA, the efficacy of our subunit vaccine was comparable to that conferred by the inactivated trivalent whole-virus vaccine (Re-6-7-8).

A broad-spectrum protective immune response, which is defined as protection against different H5 clades viruses, has been previously reported in ferret and murine models used for the development of vaccines for humans [18,40,41]. In addition, the M2 ectodomain (M2e) has also been evaluated as a broad-spectrum influenza vaccine candidate in chickens [42]. Nevertheless, its clinical application was restricted due to the lack of an appropriate method to evaluate the immune response elicited by the M2e antigens. The HA stalk region has also been used as a target antigen for the development of a universal vaccine for broad-spectrum protection [43].

HI or virus neutralization (VN) activity is generally considered as an immune correlate for the efficacy of influenza vaccines. Interestingly, the HI titers against the challenge viruses were rather low in the rBacH5Con5Mut vaccination group, irrespective of the challenge virus. However, higher HI titers were observed in response to the challenge viruses after Re-6-7-8 vaccination, which can be attributed to the exact antigenic match between these viruses and the Re-6-7-8 vaccine. Spackman et al. found that the HI antibody titers are not always predictive of the protection from challenge infection or viral shedding [44]. In some cases, adequate protection could be observed in the presence of low HI titers [45]. On the other hand, some studies have also reported a clear correlation between HI titer and the level of protection [46]. The exact mechanism underlying the broad-spectrum protective effects of rBacH5Con5Mut is not fully understood. One plausible explanation could be that the exposed epitopes of the recombinant HA are readily recognized by the immune system, resulting in an appropriate broad-spectrum immune response.

Compared to HI titers, the neutralizing antibody titers more closely reflect the ability of the antibodies to inhibit viral replication and propagation [47]. In our study, the H5N1 pseudovirus was used to evaluate the presence of neutralizing antibodies. The correlation of measurements performed using this assay was very similar with microneutralization assays using live virus [48,49]. The pVN test results showed that the pVN titer is also very low in rBacH5Con5Mut vaccination group although it can provide 100% protection. This result reinforces the complexity of the immune response required to provide full protection after challenge with antigenic and genetically diverse, rapid, and systemic replicating HPAIVs. Similar results with a lack of correlation of the neutralizing antibody data with protection have been reported in humans against H5N1 infection [50,51]. Recently, it is also found that the HA-binding non-neutralizing antibodies, but not HI or VN antibodies, contribute to the protection against H7N9 infection in chickens and total binding antibody detection can be used as a key analytical method to correlate with protection [52]. This may warrant the necessity for a different assay to evaluate the potency of this vaccine, such as the measurement of an H5-specific IgG antibody. The presence of heterosubtypic protection in the absence of strong neutralizing antibodies also indicates the possibility of cell-mediated immune (CMI) responses in providing cross-protection.

In our study, we found that the HI titer of unvaccinated controls is also about 3log2 against rBacH5 antigens. However, the HI titer against the challenge virus or the pVN titer of the unvaccinated chickens were both negative. It has been reported that the presence of non-specific inhibitors in chicken sera can affect the HI result. We hypothesize that this kind of non-specific inhibitor had an impact on the binding of rBacH5 with red blood cells which induced the HI titer of unvaccinated chickens of about 3log2. Thus, in future studies we can use either 20% kaolin or a receptor-destroying enzyme to eliminate the non-specific inhibitors to test the HI titer when using rBacH5 as the standard antigen.

We also explored the feasibility of the vaccination of one-day-old chickens because hatchery vaccination is important in the field, and it has several advantages. First, commercially used vaccination equipment designed for one-day-old chickens can be used directly in the hatchery. Second, given that there are fewer hatcheries compared to poultry farms, vaccination programs in hatcheries can be more easily conducted, controlled, and centralized. Finally, hatchery vaccination allows early onset of immunity in the chickens. The HVT-vectored vaccine [53] and fowlpox-vectored vaccine [54] have been successfully used in hatcheries. In our study, the rBacH5Con5Mut vaccine was able to confer full protection against the H5N1 virus challenge when administered to one-day-old SPF chickens, indicating that it has the potential to be employed in hatcheries as well. Nevertheless, it remains to be ascertained whether the rBacH5Con5Mut vaccine can be effective in one-day-old commercial chickens with maternally derived antibodies. The impact of the maternally derived antibodies on efficacy needs to be further evaluated.

In conclusion, a single dose vaccination of the rBacH5Con5Mut vaccine can provide broad-spectrum protection against the lethal challenge of three different clades of H5 HPAIV, efficiently prevented or reduced virus shedding. The vaccine was also efficacious when applied in one-day-old SPF chickens, indicating its potential for mass vaccination in hatcheries. In this study, the efficacy of rBacH5Con5Mut vaccine was only evaluted on SPF chickens, and the protection and vaccination program in commercial maternally derived antibody-positive chickens should be further investigated. Furthermore, as a subunit vaccine produced in the baculovirus expression system [27], the ability of the rBacH5Con5Mut vaccine to differentiate infected from vaccinated animals should be investigated. Overall, this rationally designed vaccine candidate appears to fulfill the criteria for a broad protective H5 avian influenza vaccine and could be a useful tool in combating avian influenza virus outbreaks in the future.

## Figures and Tables

**Figure 1 vaccines-12-00932-f001:**
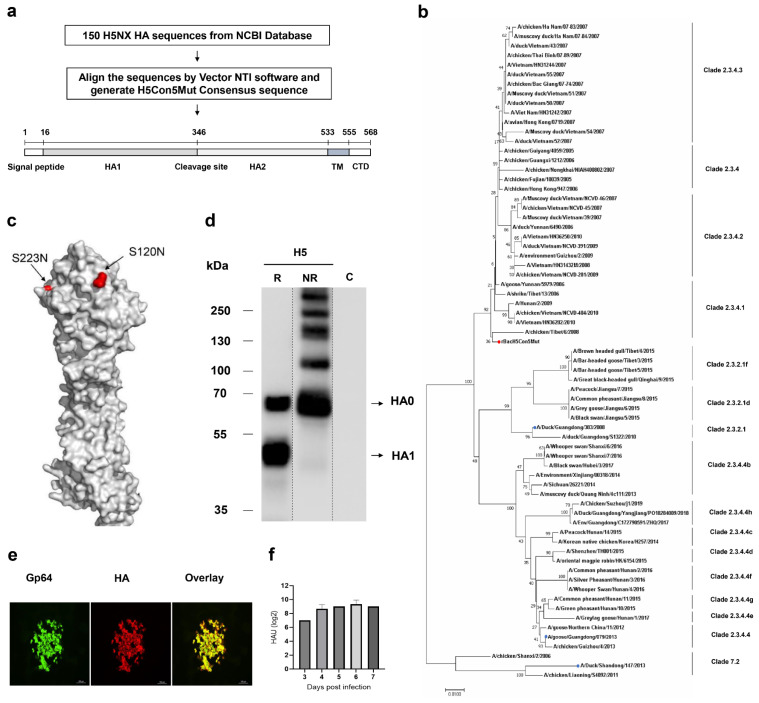
Design and characterization of rBacH5Con5Munt recombinant virus. (**a**) A total of 150 H5NX HA sequences were obtained from the NCBI database and aligned by Vector NTI software. A consensus H5NX HA sequence was generated and named as H5Con5Mut. (**b**) Phylogenetic analysis of HA sequences of H5Nx isolated from poultry, the location of consensus H5Con5Mut HA, and the challenge virus used in this study are shown in the tree. (**c**) The 3D structure model based on consensus HA protein is shown and the locations of the S120N and S223N mutations are highlighted in red on the structure. (**d**) H5 HA expression from rBacH5Con5Mut with (R) or without (NR) 10× reducing agent were confirmed by WB. The SF+ cells infected with wild-type baculovirus were set as a mock control indicated as C. (**e**) Dual IFA staining of H5 HA and Gp64 by corresponding antibodies. (**f**) Expression of H5 HA was confirmed by HA assay.

**Figure 2 vaccines-12-00932-f002:**
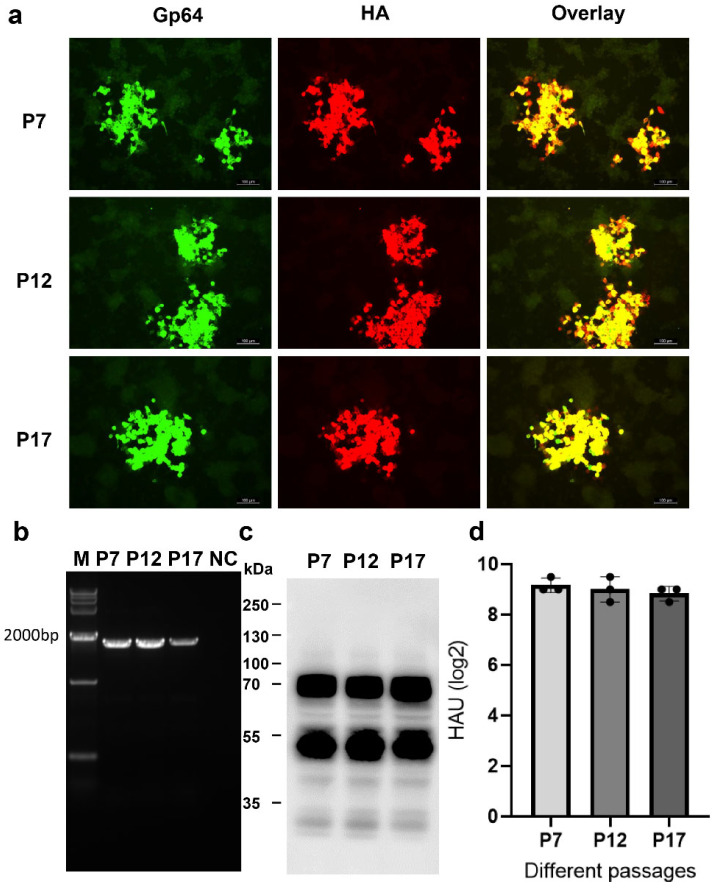
Genetic stability of HA in rBacH5Con5Mut. The rBacH5Con5Mut virus was continuously passaged in SF+ cells from P7 to P17. (**a**) Dual IFA staining of H5 HA and Gp64 protein of rBacH5Con5Mut P7, P12, and P17 viruses infected in Sf9 cells. (**b**) The presence of the H5 HA gene in the recombinant baculovirus from different passages was confirmed by PCR (~1800 bp). (**c**,**d**) H5 HA expression from rBacH5Con5Mut P7, P12, and P17 viruses was confirmed by Western blot and HA assay.

**Figure 3 vaccines-12-00932-f003:**
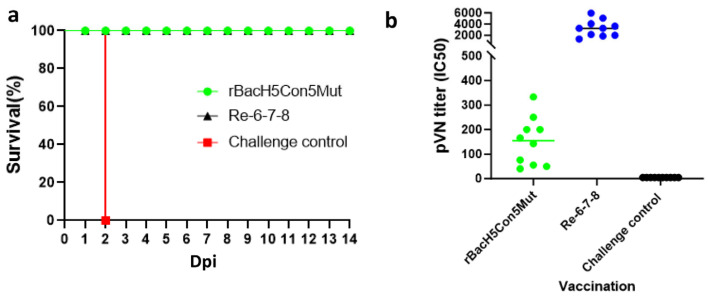
Survival rate of the chickens intranasally challenged with clade 2.3.4.4 H5N1 HPAI and pVN antibody titers at 21 dpv. Three-week-old SPF chickens were randomly divided into three groups: 10 chickens were immunized with rBacH5Con5Mut, 10 chickens were immunized with the inactivated whole-virus Re-6-7-8 vaccine, and 10 chickens were injected with PBS (challenge control group). At 21 dpv, the chickens were infected with (**a**) clade 2.3.4.4 H5N1 HPAI viruses. Clinical signs were observed for 14 days. (**b**) pVN titers of the chicken serum were tested in 2.3.4.4 challenging study using 14079-pseudovirus.

**Figure 4 vaccines-12-00932-f004:**
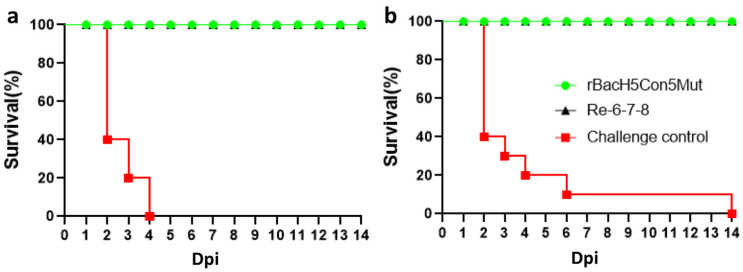
Survival rate of the chickens intranasally challenged with clade 2.3.2.1 and clade 7.2 H5 HPAI. Three-week-old SPF chickens were randomly divided into three groups: 10 chickens were immunized with rBacH5Con5Mut, 10 chickens were immunized with the inactivated whole-virus Re-6-7-8 vaccine, and 10 chickens were injected with PBS (challenge control group). At 21 dpv, the chickens were infected with (**a**) clade 2.3.2.1 H5N2 HPAI and (**b**) clade 7.2 H5N2 HPAI viruses. Clinical signs were observed for 14 days.

**Figure 5 vaccines-12-00932-f005:**
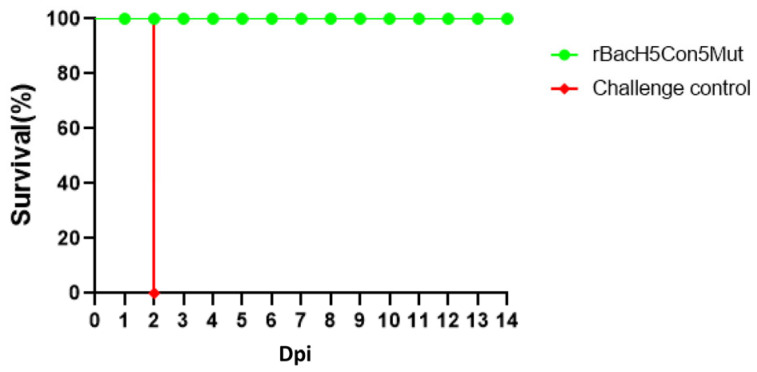
Survival rate of one-day-old SPF chickens challenged with the highly pathogenic H5 avian influenza virus. Two groups of one-day-old SPF chickens were immunized with the rBacH5Con5Mut vaccine or PBS (control group), respectively. At 21 dpv, the chickens were infected with clade 2.3.4.4 H5N1 HPAIV. Clinical signs were observed for 14 days.

**Table 1 vaccines-12-00932-t001:** Protective efficacy of the rBacH5Con5Mut vaccine against different highly pathogenic H5 AI viruses in 21-day-old chickens.

Experiment	Vaccination	Challenge Virus	Clade	HI Titer with Different Antigens (log2) ^a^		Survival/ Total	Virus Shedding from Chickens at Five Days Post-Challenge	
				rBacH5	CV ^b^		Oropharyngeal	Cloacal
1	rBacH5Con5Mut	A/Goose/Guangdong/079/2013 (H5N1, 14079)	2.3.4.4	9.1	3.6	10/10	0/10	0/10
	Re6-7-8			6.5	8	10/10	0/10	0/10
	Control			3.3	0	0/10	N/a ^c^	N/a
2	rBacH5Con5Mut	A/Duck/Guangdong/383/2008 (H5N1, 383)	2.3.2.1	8.0	2.1	10/10	0/10	0/10
	Re6-7-8			5.7	5.8	10/10	0/10	0/10
	Control			3.2	0	0/10	N/a	N/a
3	rBacH5Con5Mut	A/Duck/Shandong/147/2013 (H5N2, 147)	7.2	8.7	1.6	10/10	1/10	6/10
	Re6-7-8			6.2	6.3	10/10	1/10	0/10
	Control			3.3	0	0/10	2/2	2/2

^a^ The HI titers were measured using the homologous rBacH5Con5Mut virus or the challenge viruses with the antisera collected three weeks after vaccination. ^b^ Challenge virus. ^c^ All of the chickens in the group died before day 5.

**Table 2 vaccines-12-00932-t002:** Protective efficacy of the rBacH5Con5Mut vaccine against highly pathogenic H5 AIV strains in one-day-old chickens.

Vaccination	Challenge Virus	Clade	Days of Age	HI Titer with Different Antigens (log2) ^a^		Survival/ Total	Virus Shedding from Chickens at Five Days Post-Challenge	
				rBacH5	CV ^b^		Oropharyngeal	Cloacal
rBacH5Con5Mut	A/Goose/Guangdong/079/2013 (H5N1, 14079)	2.3.4.4	1	9.1	2.5	10/10	0/10	0/10
Challenge control			1	2	0	0/10	N/a ^c^	N/a

^a^ The HI titers were measured using the homologous rBacH5Con5Mut virus or the challenge viruses with the antisera collected three weeks after vaccination. ^b^ Challenge virus. ^c^ All of the chickens in the group died before day 3.

## Data Availability

The data presented in this study are available on request from the corresponding author.

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
