# Peer review of "A Rationally Designed H5 Hemagglutinin Subunit Vaccine Provides Broad-Spectrum Protection against Various H5Nx Highly Pathogenic Avian Influenza Viruses in Chickens"

_vaccines, 2024, doi:10.3390/vaccines12080932_

Round 1

Reviewer 1 Report

Comments and Suggestions for Authors

I found this study to be relevant, clearly written and easily understood,  thereby appealing to a wide variety of readers.  I only observed 2 small corrections in my review:

Line 15  The chicken sera of vaccinated chickens

Line 214  'was confirmed by western blot, Dual IFA and HA assays"  ( changed to match order of figure)

I did have an ethical concern regarding the animal welfare as the methods state that one of the observed clinical signs was death. I do understand that HPAI can act very quickly and often animals can die despite efforts to prevent it. However,  could you please confirm whether death was considered an endpoint in the experiment or were efforts made to  humanely euthanize control animals when they reached a point of being moribund before they died?  Please add this information to the methods of the animal study. 

Author Response

Comments 1: Line 15: The chicken sera of vaccinated chickens

Response 1: Thanks for your suggestion. The word of ‘chicken’ has been deleted in this manuscript (Line 31).

Comments 2: Line 214: 'was confirmed by western blot, Dual IFA and HA assays" (changed to match order of figure)

Response 2: The sentence has been changed with ‘HA expression in rBacH5Con5Mut-infected SF+ cells was confirmed by Western blot, dual IFA and HA assays’ to match the order of figure in the manuscript (Line 267-268).

Comments 3: I did have an ethical concern regarding the animal welfare as the methods state that one of the observed clinical signs was death. I do understand that HPAI can act very quickly and often animals can die despite efforts to prevent it. However, could you please confirm whether death was considered an endpoint in the experiment or were efforts made to humanely euthanize control animals when they reached a point of being moribund before they died?  Please add this information to the methods of the animal study.

Response 3: We follow strictly the IACUC guideline and put animal welfare on top of every of our clinical studies. For HPAI as you are aware, the chickens in the challenge only group die within 48hrs, with very short period of time shown with clinical symptoms thus in general requires no humanely intervein such as euthanizing. For vaccination/challenge groups, we do closely monitor animals and always prepared to humanely euthanize the animals if any sign of suffering is shown (Line 209-213).

Reviewer 2 Report

Comments and Suggestions for Authors

In the manuscript “A rationally designed H5 hemagglutinin subunit vaccine provides broad-spectrum protection against various H5Nx highly pathogenic avian influenza viruses in chickens” The authors developed a subunit vaccine based on single consensus H5 HA antigen and evaluating its efficacy against challenge. The manuscript is generally well-addressed; however, I have some comments/suggestions.

Line 76- 84; these are results that I suggest to move them to results/discussion, since introduction can include only the rational and aim of study.

Line 116: please provide how many sequences used in alignment and if possible, the accession number (like a link).

Line 123: western blot, please verify the type of samples type used for this assay. Do you mean by samples the supernatant?

Line 122: did you include controls in western blot? if so, please add them in text and also in figure legend (Fig 1d).

Line 132: the sentence “ HA: Samples for HA assay were serially diluted….. “, what the samples type used? please be specific.  also, rewrite this paragraph adding details (such as diluent PBS and references)

Line 136: the title " Animal studies for vaccine efficacy evaluation " need to be revised.

Line 137-138: this is confusing with the infection steps by Line 112-115, or this stock for formulation and clinical work. please revise to avoid confusion.

line 153: please specify how many birds used or how many treatments/groups. also, add the positive and negative control groups details such as  doses and materials that used in controls.

Line 155: “oil adjuvanted vaccine…”. please add the concentrations of proteins in this dose volume. I mean 0.3 ml have how much ug of antigens?

Line 157: “egg infectious dose 50 (EID50) of the challenge virus strain” please specify the challenge strain used in animal study.

Line 163: the sentence “virus was isolated from the swabs (see above) and grown in chicken embryos”,  please rewrite. It could be the virus was isolated from the collected swab samples and grown in SPF chicken embryo as follow. briefly, ...

Line 170: I suggest moving HA assay here after egg virus isolation to make sense unless you use other samples.

Line 173: “The antigen used for HI assay is either BEI inactivated rBacH5Con5Mut or formaldehyde inactivated challenge virus antigen” do you think both antigens work or behave similarly either inactivated by formalin or BEI? Is that any differences that should be mentioned?

Line 246: “rBacH5Con5Mut virus from P7, P12”, why did you use theses passages P7 and P12? is that rational for that? if so, please explain.

Line 255: Figure 2 legend: “was confirmed by WB” , please clarify the abbreviation of WB (western blot)

Line 286: Figure 3. Survival rate of the chickens intranasally challenged.... please add the survival rate in the material and methods for the animal experiment design.

References:

Please revise all references and follow the journal guidelines.

Comments on the Quality of English Language

English language is good however Minor editing is still required.

Author Response

Comments 1: Line 76-84; these are results that I suggest to move them to results/discussion, since introduction can include only the rational and aim of study.

Response 1: These sentences have already been deleted in the introduction part (Line: 90).

Comments 2: Line 116: please provide how many sequences used in alignment and if possible, the accession number (like a link).

Response 2: The accession number of 67 HA gene sequences analyzed in this study were included in the materials and methods session (Line: 128-139).

Comments 3: Line 123: western blot, please verify the type of samples type used for this assay. Do you mean by samples the supernatant?

Response 3: Because the insect cell is a suspension cell culture, and we took samples directly from the suspension cell culture post infection, thus, the sample for Western blot included both the culture supernatant and cells in the culture to confirm the H5HA expression (Line: 144-146).

Comments 4: Line 122: did you include controls in western blot? if so, please add them in text and also in figure legend (Fig 1d).

Response 4: Yes, the mock control was included in this western blot experiment. The SF+ cells infected with wild type baculovirus was used as the mock control. The detailed information has been added in the text (Line: 274-276) and figure legend (Line: 292-293).   

Comments 5: Line 132: the sentence “HA: Samples for HA assay were serially diluted….. “, what the samples type used? please be specific.  also, rewrite this paragraph adding details (such as diluent PBS and references)

Response 5: rBacH5Con5Mut infected SF+ cell suspension or allantoic fluid samples (25 ul) were serially diluted 2-fold by phosphate buffered saline (PBS) (Gibco, Waltham, MA, USA) from 1:2 to 1:2048 in 96 well V-bottom plates. In each well, 25 ul PBS was added in a 25 ul volume of the diluted samples, and then 25 ul 1% chicken red blood cells (RBCs) were added to each well. The plates were gently mixed and incubated at room temperature for 45 min. The highest dilution that showed complete hemagglutination was recorded as the HA titer for individual sample (Ref: Hemagglutination Assay for Influenza Virus)” (Line: 172-178).

Comments 6: Line 136: the title " Animal studies for vaccine efficacy evaluation " need to be revised.

Response 6: The title has been revised with “Preparation of the vaccines” (Line: 179).

Comments 7: Line 137-138: this is confusing with the infection steps by Line 112-115, or this stock for formulation and clinical work. please revise to avoid confusion.

Response 7: For the Line 112-115, it described the production of virus stock and for line 137-138, it described the process for vaccine antigen production. The MOI is different for these two experiments. To avoid confusion, the related description for production of virus stock has been added in the “Cells and viruses” part in materials and methods (Line: 107-109).

Comments 8: Line 153: please specify how many birds used or how many treatments/groups. also, add the positive and negative control groups details such as doses and materials that used in controls.

Response 8: The detailed description has been changed with “Groups of 10 twenty one-day-old or one-day-old White Leghorn SPF chickens were injected subcutaneously with 0.3 ml of oil adjuvanted vaccines containing 500 HAU of HA antigens or 0.3 ml of commercial trivalent inactivated whole virus-based mineral oil emulsion Re-6-7-8 vaccine. Groups of 10 chickens were injected subcutaneously with 0.3 ml of phosphate-buffered saline (PBS) as a control.” (Line: 197-201)

Comments 9: Line 155: “oil adjuvanted vaccine…”. please add the concentrations of proteins in this dose volume. I mean 0.3 ml have how much ug of antigens?

Response 9: It is a good suggestion. One dose is 0.3 ml containing 500 HAU of HA antigens. The sentence has been further modified as shown in Line 198.

Comments 10: Line 157: “egg infectious dose 50 (EID50) of the challenge virus strain” please specify the challenge strain used in animal study.

Response 10: The challenge strains used in the animal studies including A/Goose/Guangdong/079/2013 (H5N1), A/Duck/Guangdong/383/2008 (H5N1) and A/Duck/Shandong/147/2013 (H5N2). The information has been added in the materials and methods (Line: 203-205).

Comments 11: Line 163: the sentence “virus was isolated from the swabs (see above) and grown in chicken embryos”,  please rewrite. It could be the virus was isolated from the collected swab samples and grown in SPF chicken embryo as follow. briefly, ...

Response 11: Thanks for the revision. This sentence has been changed with “to assess virus shedding after challenge, the collected cotton swab samples were processed and inoculated in SPF chicken embryos for virus isolation.” (Line: 218-219).

Comments 12: Line 170: I suggest moving HA assay here after egg virus isolation to make sense unless you use other samples.

Response 12: Thanks for the suggestion. Not only the HA titer of the sample of collected allantoic fluid was tested, the HA titer of rBacH5Con5Mut infected SF+ cell suspension samples was also tested in the first result. The order of HA assay has not been changed but the related description of HA assay has been further optimized as shown in Line: 171-178.

Comments 13: Line 173: “The antigen used for HI assay is either BEI inactivated rBacH5Con5Mut or formaldehyde inactivated challenge virus antigen” do you think both antigens work or behave similarly either inactivated by formalin or BEI? Is that any differences that should be mentioned?

Response 13: We did preliminary evaluated the HA and HI assay in the past and compared the antigens prepared by BEI inactivated rBacH5 and formaldehyde inactivated rBacH5, we found no obvious difference regarding the HI test result. In addition, the same conclusion was reported by another group (Reference: Evaluation of different methods of inactivation of Newcastle disease virus and avian influenza virus in egg fluids and serum).

Comments 14: Line 246: “rBacH5Con5Mut virus from P7, P12”, why did you use theses passages P7 and P12? is that rational for that? if so, please explain.

Response 14: Based on the regulatory requirement for the purpose of product registration, genetic stability needs to be demonstrated for the defined master virus seed passage (P7 for rBacH5Con5Mut), and passages after 5 and 10 more additional cell culture passages. Thus we tested P7, P12 and P17 passages of the virus.  

Comments 15: Line 255: Figure 2 legend: “was confirmed by WB”, please clarify the abbreviation of WB (western blot)

Response 15: The abbreviation of WB was deleted and the sentence is now revised as: "was confirmed by Western blot” (Line: 312).

Comments 16: Line 286: Figure 3. Survival rate of the chickens intranasally challenged.... please add the survival rate in the material and methods for the animal experiment design.

Response 16: The description of survival rate with ”The survival rate is calculated by the number of surviving chickens divided by the number of total chickens over 14 days after challenge. The survival rate curves were generated with GraphPad Prism 10.” has been added in the immunization and challenge part in material and methods (Line: 213-216).

Reviewer 3 Report

Comments and Suggestions for Authors

Dear Authors,

A well-done and well-written work. The authors have developed an insect-cell-derived single consensus H5 HA antigen-based subunit vaccine that provides satisfactory protection of chicken against lethal challenge of different clades of the H5 highly pathogenic influenza viruses. The Baculovirus vector system was used for recombinant virus production and HA/H5 influenza virus protein expression. The constructed consensus sequence of the HA/H5 influenza virus gene is not presented in the article (perhaps this is know-how), but all steps of the work, appropriate reagents, methods, and commercial kits are described in the article.

Applied approach and created vaccine are very actual for China, where different clades of H5 highly pathogenic influenza virus have been circulating in poultry. This study will be useful to other researchers working in the field of vaccine development.

            However, there are some remarks concerning the text.

1. Sometimes the title of subsection does not coincide to content, or this subsection contains incomplete information. For example, I did not find the description of cells in the section ‘Cells and viruses’ (Lines 93-97). You say about influenza viruses in this section. The next section describes recombinant baculovirus. Therefore, it may be better to entitle the latter as ‘Antigen design and recombinant baculovirus production’.

 Similar remark is for the title ‘Animal studies for vaccine efficacy evaluation’. (Line 136)

2. The images shown in Fig.1d and Fig.2b,c are clearly described in the main text. Although supplied original images (vaccines-3124311-original-images-1.pdf) were useful for me, in my opinion, this file is not necessary for final version of the article.

3. Lines 36-39.

An order of references is disturbed in this part of the text. Reference 9 appears before references 7 and 8.

4. Line 123. What kind of components are included in loading buffer for western blot samples? What did you use as a reducing reagent?

5. L132, 174 What kind of saline was used for dilution in Hemagglutination assay (Line 132) and Hemagglutination inhibition (HI) assay (Line 174)?

6. L139, 173   What does abbreviation BEI mean? 

7. Lines 221, 236, 252. What is a gp64? Is it a protein of recombinant baculovirus? Please, indicate it anywhere. Different writing of this protein occurs in the text – gp64 and Gp64. Uniform writing is needed. 

With best wishes,

your reviewer

Author Response

Comments 1: Sometimes the title of subsection does not coincide to content, or this subsection contains incomplete information. For example, I did not find the description of cells in the section ‘Cells and viruses’ (Lines 93-97). You say about influenza viruses in this section. The next section describes recombinant baculovirus. Therefore, it may be better to entitle the latter as ‘Antigen design and recombinant baculovirus production’. Similar remark is for the title ‘Animal studies for vaccine efficacy evaluation’. (Line 136)

Response 1: Thank you for your kindly reminder. Some cells related information has been added in “Cells and viruses” part (Line: 99-113). The title of next section had been changed with “Antigen design and recombinant baculovirus production” in materials and methods (Line: 115). What’s more, the sentence of ‘Animal studies for vaccine efficacy evaluation’ has been changed with ‘Preparation of the vaccines’ (Line: 179).

Comments 2: The images shown in Fig.1d and Fig.2b,c are clearly described in the main text. Although supplied original images (vaccines-3124311-original-images-1.pdf) were useful for me, in my opinion, this file is not necessary for final version of the article.

Response 2: I will align with the editor to make sure the original images (vaccines-3124311-original-images-1.pdf) will not be in the final version of the article.

Comments 3: Lines 36-39. An order of references is disturbed in this part of the text. Reference 9 appears before references 7 and 8.

Response 3: The order of these references has been double checked and updated (Line: 52-55).

Comments 4: Line 123. What kind of components are included in loading buffer for western blot samples? What did you use as a reducing reagent?

Response 4: The loading buffer used in Western blot experiment contained lithium dodecyl sulfate (LDS) at pH 8.5 with SERVA Blue G250 and phenol red. The reducing reagent is 500mM DTT. The detailed information has been added in materials and methods of western blot part (Line: 146-150).

Comments 5: L132, 174 What kind of saline was used for dilution in Hemagglutination assay (Line 132) and Hemagglutination inhibition (HI) assay (Line 174)?

Response 5: Phosphate buffered saline (PBS) was used for antigen or serum dilution in Hemagglutination (HA) assay and Hemagglutination inhibition (HI) assay. The detailed information has been added in the materials and methods (Line: 171-178 and 227-234).

Comments 6: L139, 173   What does abbreviation BEI mean? 

Response 6: BEI stands for Binary Ethylenimine. The full name has been added in the “Preparation of the vaccines” part (Line: 182-183).

Comments 7: Lines 221, 236, 252. What is a gp64? Is it a protein of recombinant baculovirus? Please, indicate it anywhere. Different writing of this protein occurs in the text – gp64 and Gp64. Uniform writing is needed. 

Response 7: Gp64 stands for glycoprotein 64. It is the envelope glycoprotein of baculovirus which is an essential component for virus budding and is necessary for efficient virion assembly. This description has been added in the results part (Line: 159-160). The writing had been uniformed with Gp64.

Reviewer 4 Report

Comments and Suggestions for Authors

In the reviewed manuscript, subunit H5 avian influenza vaccine based on consensus H5 HA sequence was developed. High immunogenicity and protective  efficacy in chickens of 21 and 1 day old were demonstrated. Single immunization provide cross-protection against lethal challenge of three heterologous subclades  (2.3.4.4, 2.3.2.1 and 7.2) of H5 HPAI viruses. The methods and results are described in detail and are reproducible.

Note:

1)      It would be a good to decipher the following abbreviations when they are first mentioned:

Line 139    BEI (final concentration of 10 mM)”

Line 221    “Moreover, the expression of Gp64

Line 292      pVN titers of the chicken serum”

Line 422        HI or VN activity

2)      Figure 3.       The image did not fit in the frame

3)      It would be good to discuss why the sera of control chickens give a signal in HI test against the rBacH5 antigen

4)      Line 396     “In contrast, specific virus strain sequence-based subunit vaccines provide only strain-restricted immunity”.     A reference is needed here

5)      Line 406      “serine at amino acid residue 223 was substituted with asparagine in order to enhance the antigenicity of the vaccine candidate in chickens”     Substitution of S223N also was detected in human isolates

Author Response

Comments 1: It would be a good to decipher the following abbreviations when they are first mentioned:

Line 139  “BEI (final concentration of 10 mM)”

Line 221  “Moreover, the expression of Gp64”

Line 292  “pVN titers of the chicken serum”

Line 422  “HI or VN activity”

    Response 1: The abbreviation of Binary Ethylenimine (BEI) (Line: 182-183), glycoprotein 64 (Gp64) (Line: 159-160), pSeudovirus neutralization (pVN) (Line: 249), virus neutralization (VN) (Line: 484) has been added in the manuscript.

Comments 2: Figure 3. The image did not fit in the frame

Response 2: Thank you for kind reminder. The size of the image has been modified.

Comments 3: It would be good to discuss why the sera of control chickens give a signal in HI test against the rBacH5.

Response 3: The related description has been added in the discussion part: In our study, we found that the HI titer of unvaccinated controls is about 3log2 against rBacH5 antigens. However, the HI titer against challenge virus or the pVN titer of the unvaccinated chickens were both negative. It had been reported that the presence of non-specific inhibitors in chicken sera can affect the HI result (Reference: Comparison of serum treatments to remove nonspecific inhibitors from chicken sera for the hemagglutination inhibition test with inactivated H5N1 and H9N2 avian Influenza A virus subtypes). Our hypothesis is that this kind of non-specific inhibitor might have impact on the binding of rBacH5 HA to the red blood cells, which developed the non-specific HI titer. Thus in further studies, we can use either 20% kaolin or receptor destroying enzyme to eliminate the non-specific inhibitors to test the HI titer when using rBacH5 as standard antigen (Line: 514-521).

Comments 4: Line 396 “In contrast, specific virus strain sequence-based subunit vaccines provide only strain-restricted immunity”. A reference is needed here

Response 4: The reference of “Single Dose of Consensus Hemagglutinin-Based Virus-Like Particles Vaccine Protects Chickens against Divergent H5 Subtype Influenza Viruses” has been added after this sentence (Line: 459).

Comments 5: Line 406  “serine at amino acid residue 223 was substituted with asparagine in order to enhance the antigenicity of the vaccine candidate in chickens”  Substitution of S223N also was detected in human isolates

Response 5: Yes, you are right. The substitution of S223N was also detected in human isolates. As reported in the publication of “Role of specific hemagglutinin amino acids in the immunogenicity and protection of H5N1 influenza virus vaccines”, they found that the 223 position is located in 220-loop of the receptor binding domain. S223N can stimulate higher HI antibody titer in ferret which might because this substitution can increase the antibody-antigen binding which might can increase its antigenicity.